# SQ-CBF: Signed Distance Functions for Numerically Stable Superquadric-Based Safety Filtering

Haocheng Zhao*, Lukas Brunke*, Oliver Lagerquist, Siqi Zhou, and Angela P. Schoellig

*Abstract*—**Safety filters based on control barrier functions (CBFs) provide an effective mechanism for enforcing collision avoidance in robot manipulation, but their performance critically depends on the underlying geometric representation of the environment. This work presents a geometry-aware safety filter that models robot and environment geometries using expressive superquadrics (SQs). Our approach yields a numerically stable, real-time-capable CBF safety filter, SQ-CBF, by leveraging signed–distance functions and deriving a closed-form gradient expression. Simulation and real-world experiments show that our proposed SQ-CBF achieves stable collision avoidance while improving teleoperation efficiency in highly constrained or dynamic scenes.**

## I. INTRODUCTION

Safety is a fundamental requirement in robot control, particularly for systems operating in close proximity to obstacles or humans [1]. Safety filters based on control barrier functions (CBFs) are a practical way to enforce collision avoidance online [2] [3], but their effectiveness depends on the underlying collision model. Common primitive-based approximations (e.g., spheres or boxes) are computationally efficient yet often too inaccurate or overly conservative in cluttered scenes [4], degrading both safety and task performance. In this work, we enable expressive and numerically stable geometry-aware safety filtering by modeling robot and scene geometry with compact superquadrics [5] [6] and executing the distance query through efficient Gilbert-Johnson-Keerthi algorithm [7]. Experiments in simulation and on a real robot demonstrate reliable collision-free teleoperation in tight, cluttered, and dynamic environments, while reducing unnecessary corrective motions and improving efficiency.

## II. METHODOLOGY

### A. Robot and Environment Representation

Both the robot and the environment are represented by SQs, which provide a compact yet expressive approximation of complex scenes. The robot is modeled as a collection

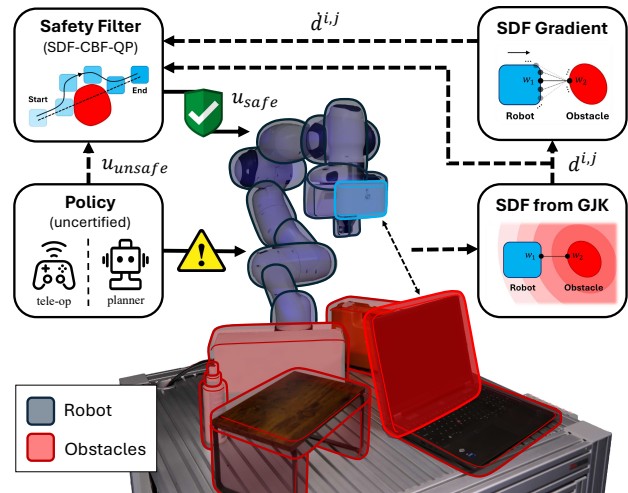

Fig. 1: Overview of the proposed SQ-CBF framework for geometry-aware safety filtering. The framework represents both the robot (blue) and the obstacles (red) using compact superquadrics and enforces collision avoidance through an SDF-based CBF. Given an unverified control command (e.g., from teleoperation), the safety filter modifies the command online by solving a CBF-QP, intervening only when safety is at risk. A video demonstrating the safety filter's performance can be found here: http://tiny.cc/sq-cbf.

of SQs rigidly attached to its kinematic chain, denoted by $\mathcal{S}_{\mathrm{sq}}^i(\boldsymbol{q})$ for $i \in \mathcal{I}_{\mathrm{sq}}^{\mathrm{rob}} \triangleq \{1, \ldots, p\}$. The $i$-th robot SQ is attached to link $\ell(i) \in \{1, \ldots, n\}$. It is defined by a fixed local transformation relative to the link frame, parameterized by a position offset $\boldsymbol{p}_{\ell(i) \to i}$ and an orientation offset $\boldsymbol{\phi}_{\ell(i) \to i}$. As a result, the 6D pose of each robot SQ is uniquely determined by the robot's forward kinematics. Meanwhile, obstacles in the environment are modeled as SQs with free 6D poses $\mathcal{S}_{\mathrm{sq}}^j(\boldsymbol{x}_{\mathrm{obs}}^j)$ for $j \in \mathcal{I}_{\mathrm{sq}}^{\mathrm{obs}} \triangleq \{1, \ldots, q\}$. In practice, such obstacle SQs can be fitted based on the RGB-D point clouds from perception. Note that in this work, all SQs are restricted to the convex parameter regime (i.e., $e_{1,2} \leq 2$), ensuring valid and unique signed distance and gradient computation between shapes.

### B. Signed Distance Function as CBF

Consider $\mathcal{S}_{\mathrm{sq}}^i(\boldsymbol{q})$ attached to the robot and $\mathcal{S}_{\mathrm{sq}}^j(\boldsymbol{x}_{\mathrm{obs}}^j)$ as a part of an obstacle, the SDF between these two convex shapes can be formulated as a constrained minimization problem over their Minkowski difference:

$$d^{i,j}(\boldsymbol{q}, \boldsymbol{x}_{\mathrm{obs}}^j) = \begin{cases} \displaystyle\min_{\Delta\boldsymbol{p} \in \partial\mathcal{M}^{i,j}} \|\Delta\boldsymbol{p}\|, & \boldsymbol{0} \notin \mathcal{M}^{i,j}, \\ -\displaystyle\min_{\Delta\boldsymbol{p} \in \partial\mathcal{M}^{i,j}} \|\Delta\boldsymbol{p}\|, & \boldsymbol{0} \in \mathcal{M}^{i,j}, \end{cases}$$

where $\mathcal{M}^{i,j} = \{ \Delta\boldsymbol{p} = \boldsymbol{p}_1 - \boldsymbol{p}_2 \mid \boldsymbol{p}_1 \in \mathcal{S}_{\mathrm{sq}}^i; \boldsymbol{p}_2 \in \mathcal{S}_{\mathrm{sq}}^j \}$ is the Minkowski difference. Geometrically, $d^{i,j} > 0$ indicates separation between the two shapes, $d^{i,j} = 0$ corresponds to tangential contact, and $d^{i,j} < 0$ denotes interpenetration.

*Equal contribution.

Haocheng Zhao is with the Learning Systems and Robotics Lab, Technical University of Munich, 80333 Munich, Germany. Email: haocheng.zhao@tum.de

Lukas Brunke and Angela P. Schoellig are with the Learning Systems and Robotics Lab, Technical University of Munich, 80333 Munich, Germany, also with the University of Toronto Institute for Aerospace Studies, North York, ON M3H 5T6, Canada, and also with the Vector Institute for Artificial Intelligence, Toronto, ON M5G 0C6, Canada. Emails: {lukas.brunke, angela.schoellig}@tum.de

Oliver Lagerquist is with the Learning Systems and Robotics Lab, Technical University of Munich, 80333 Munich, Germany, and also with the University of Toronto Institute for Aerospace Studies, North York, ON M3H 5T6, Canada. Email: oliver.lagerquist@mail.utoronto.ca

Siqi Zhou is with the Learning Systems and Robotics Lab, Technical University of Munich, 80333 Munich, Germany, and also with Simon Fraser University, Burnaby, BC V5A 1S6, Canada. Email: siqi@sfu.ca

A CBF candidate based on $d^{i,j}$ can be formulated as

$$h_\delta^{i,j}(\boldsymbol{q}, \boldsymbol{x}_{\mathrm{obs}}^j) = d^{i,j}(\boldsymbol{q}, \boldsymbol{x}_{\mathrm{obs}}^j) - \epsilon_\delta,$$

where $\epsilon_\delta > 0$ denotes a prescribed safety margin. The associated CBF condition enforces a minimal safety distance by maintaining $h_\delta^{i,j} > 0$, which ensures that the two SQs remain collision-free at all times. For practical and efficient evaluation of the SDF, we employ the GJK algorithm and EPA [7,8], which return the minimum distance, or penetration depth, between two polytopes. We obtain a polytopic approximation for each SQ $\mathcal{S}_{\mathrm{sq}}$ by discretizing its surface into a dense meshgrid of points using uniform sampling [9]. This enables fast, robust distance evaluation, which is subsequently integrated into the CBF constraints.

### C. Gradient of Signed Distance

We leverage the geometric properties of support function and separation vector to derive the SDF gradient w.r.t. the relative motion between two SQs in closed form. Consider two convex shapes $\mathcal{A}_i$ for $i \in \{1, 2\}$ defined in their local frames $\mathcal{F}_i$. Let $\sigma_\mathcal{A}(\cdot)$ denote the support function of set $\mathcal{A}$ and $\nabla\sigma_\mathcal{A}(\cdot)$ its gradient, which returns a support point on $\partial\mathcal{A}$. We assume that $\sigma_\mathcal{A}(\cdot)$ is twice continuously differentiable. Let $\boldsymbol{x} = [\boldsymbol{t}^\top, \boldsymbol{\phi}^\top]^\top \in \mathbb{R}^6$ parameterize the relative translation and axis-angle of frame $\mathcal{F}_2$ with respect to $\mathcal{F}_1$, with the corresponding rotation $\boldsymbol{R}(\boldsymbol{\phi}) \in SO(3)$.

The optimal separation vector $\Delta\boldsymbol{p}^*$ expressed in $\mathcal{F}_1$ is governed by the stationarity condition $\boldsymbol{f}(\Delta\boldsymbol{p}^*, \boldsymbol{x}) = \boldsymbol{0}$ with

$$\boldsymbol{f}(\Delta\boldsymbol{p}^*, \boldsymbol{x}) = \Delta\boldsymbol{p}^* - \boldsymbol{p}_1^* + (\boldsymbol{t} + \boldsymbol{R}\,\boldsymbol{p}_2^*),$$

where $\boldsymbol{p}_1^*$ and $\boldsymbol{p}_2^*$ denote the witness points correspond to $\Delta\boldsymbol{p}^*$ on $\partial\mathcal{A}_1$ and $\partial\mathcal{A}_2$ expressed in local frame:

$$\boldsymbol{p}_1^* = \nabla\sigma_{\mathcal{A}_1}(-\Delta\boldsymbol{p}^*), \quad \boldsymbol{p}_2^* = \nabla\sigma_{\mathcal{A}_2}(\boldsymbol{R}^\top \Delta\boldsymbol{p}^*).$$

The gradient of $\Delta\boldsymbol{p}^*$ w.r.t. $\boldsymbol{x}$ can be derived with the implicit function theorem:

$$\frac{\partial\Delta\boldsymbol{p}^*}{\partial\boldsymbol{x}} = -\left(\frac{\partial\boldsymbol{f}}{\partial\Delta\boldsymbol{p}^*}\right)^{-1}\frac{\partial\boldsymbol{f}}{\partial\boldsymbol{x}}. \tag{1}$$

The Jacobian $\partial\boldsymbol{f}/\partial\Delta\boldsymbol{p}^*$ admits the closed form:

$$\frac{\partial\boldsymbol{f}}{\partial\Delta\boldsymbol{p}^*} = \boldsymbol{I} + \nabla^2\sigma_{\mathcal{A}_1}(-\Delta\boldsymbol{p}^*) + \boldsymbol{R}\,\nabla^2\sigma_{\mathcal{A}_2}(\boldsymbol{R}^\top\Delta\boldsymbol{p}^*)\,\boldsymbol{R}^\top, \tag{2}$$

where $\nabla^2\sigma_\mathcal{A}(\cdot)$ denotes the Hessian of the support function. The gradient $\partial\boldsymbol{f}/\partial\boldsymbol{x}$ is obtained by applying the chain rule:

$$\frac{\partial\boldsymbol{f}}{\partial\boldsymbol{x}} = \begin{bmatrix} \boldsymbol{I} & -\boldsymbol{R}\big([\boldsymbol{p}_2^*]_\times + \nabla^2\sigma_{\mathcal{A}_2}(\boldsymbol{R}^\top\Delta\boldsymbol{p}^*)[\boldsymbol{R}^\top\Delta\boldsymbol{p}^*]_\times\big)\boldsymbol{J}_r \end{bmatrix}, \tag{3}$$

where $[\cdot]_\times$ denotes the hat operator in screw theory, and $\boldsymbol{J}_r$ is the right Jacobian following the convention in [10].

Based on $\Delta\boldsymbol{p}$, the gradient of the signed distance $d$ w.r.t. the relative pose $\boldsymbol{x}$ can be expressed as

$$\boldsymbol{J}_{\boldsymbol{x}} \triangleq \frac{\partial d}{\partial\boldsymbol{x}} = \mathrm{sign}(d)\frac{\Delta\boldsymbol{p}^{*\top}}{\|\Delta\boldsymbol{p}^*\|}\frac{\partial\Delta\boldsymbol{p}^*}{\partial\boldsymbol{x}}, \tag{4}$$

where $\|\Delta\boldsymbol{p}^*\|$ denotes the norm of $\Delta\boldsymbol{p}^*$, and $\mathrm{sign}(d)$ indicates the different directions of separation or penetration.

The support function is positively homogeneous of degree one. By Euler's homogeneous function theorem, this implies

$$\nabla\sigma_\mathcal{A}(\boldsymbol{x})^\top\boldsymbol{x} = \sigma_\mathcal{A}(\boldsymbol{x}).$$

Taking the derivative with respect to $\boldsymbol{x}$ leads to

$$\nabla^2\sigma_\mathcal{A}(\boldsymbol{x})\,\boldsymbol{x} = \boldsymbol{0}. \tag{5}$$

Applying (5) to $\mathcal{A}_1$ with $\boldsymbol{x} = -\Delta\boldsymbol{p}^*$ and to $\mathcal{A}_2$ with $\boldsymbol{x} = \boldsymbol{R}^\top\Delta\boldsymbol{p}^*$ yields

$$\begin{cases} \nabla^2\sigma_{\mathcal{A}_1}(-\Delta\boldsymbol{p}^*)\,\Delta\boldsymbol{p}^* = \boldsymbol{0}, \\ \nabla^2\sigma_{\mathcal{A}_2}(\boldsymbol{R}^\top\Delta\boldsymbol{p}^*)\,\boldsymbol{R}^\top\Delta\boldsymbol{p}^* = \boldsymbol{0}. \end{cases} \tag{6}$$

Right-multiplying (2) by $\Delta\boldsymbol{p}^*$ and exploiting (6) leads to

$$\frac{\partial\boldsymbol{f}}{\partial\Delta\boldsymbol{p}^*}\Delta\boldsymbol{p}^* = \Delta\boldsymbol{p}^*,$$

which establishes that $\Delta\boldsymbol{p}^*$ is a right eigenvector of $\frac{\partial\boldsymbol{f}}{\partial\Delta\boldsymbol{p}^*}$ with eigenvalue 1. Since the Hessian terms are positive semidefinite, $\frac{\partial\boldsymbol{f}}{\partial\Delta\boldsymbol{p}^*} = \boldsymbol{I} + (\cdot)$ is positive definite and therefore invertible. Applying the inverse and transposing gives

$$\Delta\boldsymbol{p}^{*\top}\left(\frac{\partial\boldsymbol{f}}{\partial\Delta\boldsymbol{p}^*}\right)^{-1} = \Delta\boldsymbol{p}^{*\top}. \tag{7}$$

Substituting (1) into (4) and invoking (7) simplifies the gradient to

$$\boldsymbol{J}_{\boldsymbol{x}} = -\mathrm{sign}(d)\frac{\Delta\boldsymbol{p}^{*\top}}{\|\Delta\boldsymbol{p}^*\|}\frac{\partial\boldsymbol{f}}{\partial\boldsymbol{x}}.$$

Incorporating (3) and exploiting (6) to simplify the rotational component yields the final expression of SDF gradient:

$$\boldsymbol{J}_{\boldsymbol{x}} = -\mathrm{sign}(d)\frac{\Delta\boldsymbol{p}^{*\top}}{\|\Delta\boldsymbol{p}^*\|}\begin{bmatrix} \boldsymbol{I} & -\boldsymbol{R}\,[\boldsymbol{p}_2^*]_\times\,\boldsymbol{J}_r(\boldsymbol{\phi}) \end{bmatrix}, \tag{8}$$

thereby eliminating all computationally intensive Hessian terms while preserving exactness.

### D. Estimation of the CBF Derivative

With the simplified SDF gradient, the CBF time derivative can be efficiently computed in closed form. The robot SQs and their motions are parameterized by $\boldsymbol{q}$ and $\dot{\boldsymbol{q}}$, whereas each obstacle SQ $\mathcal{S}_{\mathrm{sq}}^j$ and its motion are described by the 6D pose $\boldsymbol{x}_{\mathrm{obs}}^j$ and the time derivative $\dot{\boldsymbol{x}}_{\mathrm{obs}}^j$, respectively. Using the chain rule, the SDF's time derivative is

$$\dot{d}^{i,j}(\boldsymbol{q}, \boldsymbol{x}_{\mathrm{obs}}^j) = \boldsymbol{J}_{\boldsymbol{x}^i}^{i,j}\dot{\boldsymbol{x}}_{\mathrm{rob}}^i(\boldsymbol{q}) + \boldsymbol{J}_{\boldsymbol{x}^j}^{i,j}\dot{\boldsymbol{x}}_{\mathrm{obs}}^j,$$

where SDF gradients $\boldsymbol{J}_{\boldsymbol{x}^i}^{i,j} = \frac{\partial d^{i,j}}{\partial\boldsymbol{x}_{\mathrm{rob}}^i}$ and $\boldsymbol{J}_{\boldsymbol{x}^j}^{i,j} = \frac{\partial d^{i,j}}{\partial\boldsymbol{x}_{\mathrm{obs}}^j}$ can be computed with (8). While the obstacle state $\boldsymbol{x}_{\mathrm{obs}}^j$ and its time derivative $\dot{\boldsymbol{x}}_{\mathrm{obs}}^j$ can be directly obtained from the perception pipeline, the state rates of robot SQ $\dot{\boldsymbol{x}}_{\mathrm{rob}}^i$ must be obtained from the robot's joint velocities $\dot{\boldsymbol{q}}$.

Specifically, for the $i$-th SQ attached to link $\ell(i)$, we map the joint velocities to the SQ's analytical state derivative using a two-stage kinematic transformation. First, we map the joint velocities to the spatial twist:

$$\begin{bmatrix} \boldsymbol{v}_{\ell(i)} \\ \boldsymbol{\omega}_{\ell(i)} \end{bmatrix} = \boldsymbol{J}_{\ell(i)}(\boldsymbol{q})\dot{\boldsymbol{q}} = \begin{bmatrix} \boldsymbol{J}_{\ell(i)}^v(\boldsymbol{q}) \\ \boldsymbol{J}_{\ell(i)}^\omega(\boldsymbol{q}) \end{bmatrix}\dot{\boldsymbol{q}},$$

where $\boldsymbol{J}_{\ell(i)}(\boldsymbol{q})$ denotes the geometric Jacobian up to link $\ell(i)$ [11]. Second, we transform the resulting twist into the analytical state derivative of the attached SQ:

$$\dot{\boldsymbol{x}}_{\text{rob}}^i = \boldsymbol{X}_{\ell(i)\to i} \begin{bmatrix} \boldsymbol{v}_{\ell(i)} \\ \boldsymbol{\omega}_{\ell(i)} \end{bmatrix} = \begin{bmatrix} \boldsymbol{I}_3 & -[\boldsymbol{p}_{\ell(i)\to i}]_\times \\ \boldsymbol{0}_3 & \boldsymbol{J}_l^{-1}(\boldsymbol{\phi}_{\ell(i)\to i}) \end{bmatrix} \begin{bmatrix} \boldsymbol{v}_{\ell(i)} \\ \boldsymbol{\omega}_{\ell(i)} \end{bmatrix},$$

where $[\cdot]_\times$ is the skew-symmetric operator, $\boldsymbol{X}_{\ell(i)\to i}$ represents the local transformation from link frame to SQ frame, and $\boldsymbol{J}_l(\cdot)$ is the left Jacobian of SO(3) [10].

By choosing $\boldsymbol{u} = \dot{\boldsymbol{q}}$, the SDF-CBF condition can be compactly written as

$$\boldsymbol{J}_{\delta|i}^{i,j}\boldsymbol{u} + \dot{d}_{\text{obs}}^{i,j} \geq -\alpha_\delta(h_\delta^{i,j}),$$

where $\boldsymbol{J}_{\delta|i}^{i,j} = \boldsymbol{J}_{\boldsymbol{x}^i}^{i,j}\boldsymbol{X}_{\ell(i)\to i}\boldsymbol{J}_{\ell(i)}$, $\dot{d}_{\text{obs}}^{i,j} = \boldsymbol{J}_{\boldsymbol{x}^j}^{i,j}\dot{\boldsymbol{x}}_{\text{obs}}^j$, and $\alpha_\delta$ is a class-$\mathcal{K}$ function.

### E. Safety Filter Formulation

With all the definitions above, collision avoidance and singularity avoidance are jointly enforced by solving a single quadratic program at each control cycle:

$$\boldsymbol{u}^* = \arg\min_{\boldsymbol{u}\in\mathcal{U}} \left\| \boldsymbol{J}(\boldsymbol{q})(\boldsymbol{u} - \boldsymbol{u}_{\text{cmd}}) \right\|_2^2 + \left\| \boldsymbol{u} - \boldsymbol{u}_{\text{cmd}} \right\|_2^2 \quad (9)$$

$$\text{s.t. } \boldsymbol{J}_{\delta|i}^{i,j}\boldsymbol{u} + \dot{d}_{\text{obs}}^{i,j} \geq -\alpha_\delta(h_\delta^{i,j}), \quad \forall(i,j)\in\mathcal{P}_{\text{env}},$$

$$\left(\boldsymbol{J}_{\delta|i}^{i,j} + \boldsymbol{J}_{\delta|j}^{i,j}\right)\boldsymbol{u} \geq -\alpha_\delta(h_\delta^{i,j}), \quad \forall(i,j)\in\mathcal{P}_{\text{self}},$$

where we have dropped function arguments for brevity, $\mathcal{P}_{\text{env}} = \mathcal{I}_{\text{sq}}^{\text{rob}} \times \mathcal{I}_{\text{sq}}^{\text{obs}}$, and $\mathcal{P}_{\text{self}} \subset \mathcal{I}_{\text{sq}}^{\text{rob}} \times \mathcal{I}_{\text{sq}}^{\text{rob}}$ is the set of robot SQ pairs considered for self-collision avoidance. The resulting optimization problem can be solved efficiently online using standard QP solvers, allowing the proposed safety filter to operate in real time even as the problem scales up with a large number of collision constraints.

## III. EXPERIMENTS

### A. Simulation Experiments

We evaluate the proposed safety filter in simulation on a teleoperated insertion task in which the robot must insert its end-effector into a confined container with progressively reduced clearance, creating increasingly challenging geometric conditions. Without assistance, precise teleoperation in such narrow spaces is difficult and prone to collision, as confirmed by the frequent safety violations observed in unfiltered executions in Fig. 2(b)–(c). In contrast, the proposed safety filter consistently maintains collision-free execution across all tested scenarios. At the same time, the safety filter improves task efficiency by enabling smoother and more direct motions. These results demonstrate that the proposed method provides effective online collision avoidance without introducing unnecessary conservativeness, even in highly constrained environments.

### B. Real-World Experiments

To evaluate the proposed safety filter in real-world settings, we design three representative teleoperated manipulation tasks with increasing geometric complexity: object handover in a confined workspace, object transportation through narrow passages, and manipulation in presence of dynamic obstacles. These tasks jointly assess whole-body

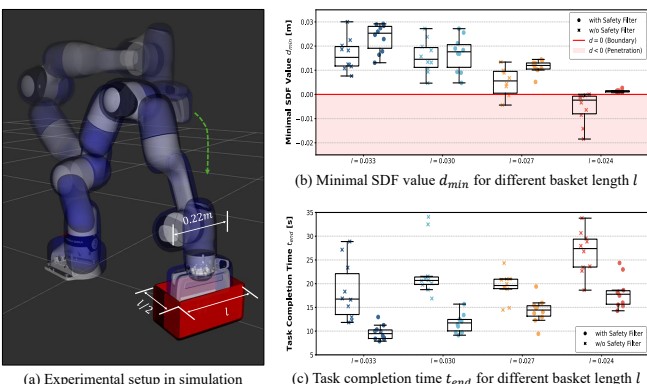

(a) Experimental setup in simulation

(b) Minimal SDF value $d_{min}$ for different basket length $l$

(c) Task completion time $t_{end}$ for different basket length $l$

Fig. 2: Simulation experiment of a teleoperated insertion task under progressively tighter geometric constraints, comparing executions with and without safety filtering and using different collision models. The results show that the proposed superquadric-based safety filter consistently prevents collisions and improves task efficiency in constrained settings.

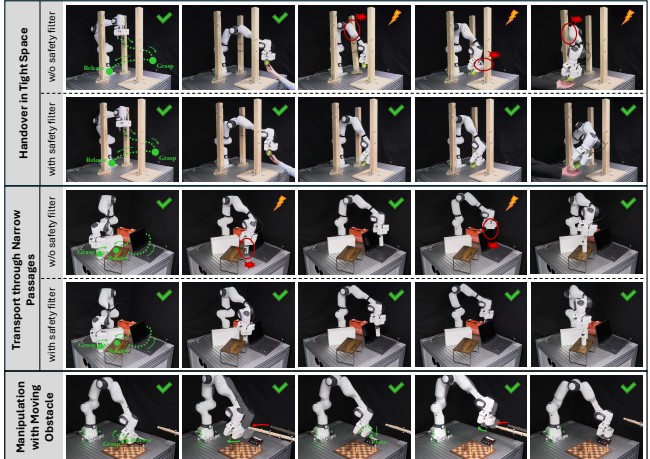

Fig. 3: Real-world teleoperated manipulation experiments demonstrating the proposed safety filter on three representative tasks. Across all tasks, the safety filter consistently enables collision-free execution with minimal intervention, despite the presence of tight geometric constraints, unstructured and cluttered obstacles, and dynamic disturbances.

collision avoidance, handling of grasped objects, and online response to moving obstacles. Figure 3 shows representative executions for each task. In all scenarios, the proposed safety filter enables collision-free task execution despite cluttered environments and dynamic disturbances. Collision avoidance is achieved without interrupting task progression or noticeably altering user commands, demonstrating that the safety filter provides robust and stable online protection during real-world teleoperation.

## IV. CONCLUSION

In this work, we presented a geometry-aware safety filter that combines expressive SQ-based collision models with an signed distance-based CBF formulation. The proposed approach bridges the gap between high-fidelity geometric modeling and reliable, real-time gradient-based safety filtering. Extensive simulations and real-world experiments demonstrate that the proposed safety filter consistently achieves collision-free execution under challenging geometric conditions, sensing noise, and dynamic disturbances, while also improving task efficiency in teleoperated manipulation by reducing unnecessary corrective motions.

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
