# OpenReview forum: "SQ-CBF: Signed Distance Functions for Numerically Stable Superquadric-Based Safety Filtering"
_IEEE.org/ICRA/2026/Workshop/Manipulation_Robustness — ICRA 2026_

### Official Review · Reviewer_VqJp · 2026-05-04
**The proposed framework SQ-CBF solves collison avoidance while achieving the task objectives in both static and dynamic environments. Although it performs exceptionally well in experiments the proposed framework lacks in theoretical analysis when measurement of obstacle 6d pose is uncertain resulting in uncertain environments.**

**Rating:** 9
**Confidence:** 4

**Review:**

The study titled ``*SQ-CBF: Signed Distance Functions for Numerically Stable Superquadric-Based Safety Filtering*" proposed an SQ-CBF framework to avoid obstacles while achieving the given task objective by leveraging the geometry of both obstacle(s) and the robot. In particular, the proposed framework solves complex task objectives while performing collision avoidance in cluttered environments. The simulation and experimental demonstration provide the efficacy of the proposed scheme. However, there is one major concern mentioned below that is not clear from the proposed solution:

1. In the proposed solution, the authors assume that the measurement of obstacle(s) position and its corresponding time derivative are known. In such a case, the robot is aware of the environment. Then, how does the robot solve the same problem in uncertain environments (which can arise from uncertainty in the measurement of obstacle position), and how does the proposed QP formulation modify?
2. Although the authors mention that the obstacle's position and its time derivative are being calculated via the perception pipeline, the uncertainty in measurements arising from the perception is not accounted for in the proposed framework. It would have been beneficial to provide robustness analysis (in measurement uncertainty) from a theoretical contribution perspective.


Moreover, the experimental demonstration in the video shows the robustness of manipulation task objectives in cluttered environments, providing a strong contribution towards collision avoidance in both dynamic and static environments. Finally, I recommend this study for possible presentation at the ICRA workshop.

---

### Decision · Program_Chairs · 2026-05-21

Accept